# Carotid Artery Disease in the Era of Biomarkers: A Pilot Study

**DOI:** 10.3390/diagnostics13040644

**Published:** 2023-02-09

**Authors:** Ettore Dinoto, Domenico Mirabella, Francesca Ferlito, Graziella Tortomasi, Davide Turchino, Salvatore Evola, Massimiliano Zingales, Emanuela Bologna, Felice Pecoraro

**Affiliations:** 1Vascular Surgery Unit, AOUP Policlinico “P. Giaccone”, 90127 Palermo, Italy; 2Department of Public Health, Vascular Surgery Unit, University Federico II of Naples, 80138 Napoli, Italy; 3Unit of Cardiology, Department of Health Promotion, Mother and Child Care, Internal Medicine and Medical Specialties (ProMISE) ‘G. D’Alessandro’, University Hospital Paolo Giaccone, University of Palermo, 90133 Palermo, Italy; 4Department of Engineering, University of Palermo, Viale delle Scienze ed. 8, 90128 Palermo, Italy; 5Department of Surgical, Oncological and Oral Sciences, University of Palermo, 90127 Palermo, Italy

**Keywords:** intima-media thickness, carotid artery disease, biomarkers, atherosclerosis, cardiovascular risk

## Abstract

The intima-media thickness (IMT) and its irregularities or ulcerations in the common carotid artery (CCA) are useful tools as sentinel biomarkers for the integrity of the cardiovascular system. Total homocysteine and lipoprotein levels are the most commonly used elements in cardiovascular risk stratification. Duplex ultrasound (DUS), associated with serum biomarkers, can be used simply to assess the degree of atherosclerotic disease and cardiovascular risk. This study highlights the role of different kinds of biomarkers, showing their usefulness and potentiality in multi-district atherosclerotic patients, especially for early diagnosis and therapy effectiveness monitoring. A retrospective analysis performed from September 2021 to August 2022, of patients with carotid artery disease, was performed. A total of 341 patients with a mean age of 53.8 years were included in the study. The outcomes showed an increased risk of stroke in patients with significative carotid artery disease, nonresponsive to therapy, monitored through a series of serum biomarkers (homocysteine, C-reactive protein, and oxidized LDL). In this reported experience, the systematic use of DUS in association with the multiple biomarkers approach was effective for the early identification of patients at higher risk of disease progression or inefficient therapy.

## 1. Introduction

Cerebral stroke is one of the most common causes of death and disability. Several studies identified the intima-media thickness (IMT) and its irregularities or ulcerations in the common carotid artery (CCA) as a sentinel biomarker for the integrity of the cardiovascular system using duplex ultrasound (DUS) [1]. At least 20% of ischemic strokes result from the rupture of an unstable plaque located at the carotid artery bifurcation [2]. The presence of lipids in the wall of large arteries may elicit an inflammatory response leading to diapedesis and an accumulation of circulating monocytes, thus forming a plaque [3]. This process is frequent in atherosclerotic disease. In daily clinical practice, it is frequently associated with a diagnosis of carotid disease in patients undergoing cardiologic or peripheral artery evaluation and vice versa. Dual antiplatelet therapy initiated within 24 h of symptom onset (TIA and minor stroke) and, for select patients with disabling ischemic stroke, thrombolysis within 4 h with possible endovascular thrombectomy plus carotid stenting, can improve functional outcomes [4,5,6]. Postmortem studies demonstrated such findings with a high correlation in these different arterial districts [7,8]. Additionally, serum biomarkers in atherosclerotic carotid artery disease have been evaluated with a particular interest in the associations of lipoprotein and inflammatory factors [9]. On this basis, the association between ultrasound biomarkers and serum biomarkers is significant. Considering the relevant role of DUS biomarkers (IMT and irregularity of carotid artery) and serum biomarkers, their correlation would play a role in identifying, more accurately, patients at risk of stroke. If confirmed, these correlations could lead to a reduction in diagnostic time and a related reduction in treatment risk and improvement of patient prognosis and quality of life. The current study aims to report our experience in patient management with any grade of carotid disease and the correlation with cardiovascular disease and biomarker findings.

## 2. Materials and Methods

This retrospective study included 341 patients presenting with CCA atherosclerosis from September 2021 to August 2022. The study included patients with different degrees of CCA atherosclerosis in the observation period. All the included patients were checked with DUS, a non-invasive technique to assess atherosclerotic disease. Symptomatic patients and presenting IMT < 1 mm were excluded from the study (Figure 1). All patients were collected and inserted into standardized piloted forms.

All the included patients gave informed consent for the procedure itself, anonymous data collection, and analysis. The study was performed in agreement with the Declaration of Helsinki, and the STROBE guidelines for reporting observational studies were followed [10]. All participants underwent baseline clinical examinations, which included medical history and physical examination. The initial diagnosis assessment included a DUS (model SSA-270A; Toshiba America Medical Systems, Tustin, Calif). The longitudinal image of the carotid artery bifurcation, the internal carotid artery and two traversal images of the CCA of all patients were acquired. The maximal CCA IMT was defined as the mean between the maximal IMT on both CCA left- and right-side images. The IMT was defined as the distance between the near and far walls. Native vessel assessment parameters for DUS included peak systolic velocity (PSV), IMT measurement, and plaque morphology evaluation. Surface characteristics were classified as smooth, irregular, ulcerated (if there was a depression of >2 mm into the media), soft (hypoechoic or isoechoic due to a prevalent component of lipids), and calcified. In patients presenting a carotid artery plaque, degree of stenosis was evaluated according to the ECST method (Figure 2) [11].

A computed tomography angiography (CT) was performed in patients presenting significative stenosis (>70%) according to DUS (Figure 3).

The biomarker analysis was performed in all patients at baseline and intervals of three months. The biomarkers analyzed were homocysteine, C-reactive protein (C-RP), and oxidized LDL. The collected variables were demographics, comorbidities, clinical data, imaging studies, and medical therapy. Renal function was estimated with the Chronic Kidney Disease Epidemiology Collaboration (CKD-EPI) [12]. The measured outcomes were the carotid artery stenosis degree, neurologic events, cardiac events, diagnosis of symptomatic and asymptomatic coronary disease, diagnosis of symptomatic and asymptomatic peripheral artery disease (PAD), and mortality rate. The exclusion criteria were the previous diagnosis of atrial fibrillation, prothrombotic state, and neurologic symptomatic lesion before the baseline. After the first DUS, the patients were divided into four groups:-BORDERLINE: Patients with IMT within the range of 1 mm to 1.5 mm. Here, considering the absence of true plaque but the presence of first modifications in the intima layer, our protocol provided a clinical evaluation with the risk factor identification (Figure 4).

-LOW GRADE and MEDIUM GRADE: Patients presenting a carotid plaque under 30% of stenosis. In these cases, a cardiology evaluation was added, and a clinical path of the main arterial districts (aorta and peripheral arteries) was concluded by DUS (Figure 5 and Figure 6).

-HIGH GRADE: Patients presenting a carotid plaque over 30% and further divided in the first step (under 50% of stenosis), the second step (stenosis between 50% and 70%), the third step (above 70%). All patients in this group underwent cardiologic assessment, with echocardiography, electrocardiogram, and DUS of peripheral arteries (Figure 7).

All patients with carotid stenosis were treated with medical therapy (statins, acetylsalicylic acid, L-Metilfolato, vitamins B2, B6, and B12) according to the European Society of Vascular Surgery (ESVS) Guidelines: Section A—prevention in patients with carotid stenosis [13]. During follow-up, every patient with a decrease in serum biomarkers and lack of plaque increase was defined as “responsive to therapy”. The assessment of hemodynamically significant stenosis was completed with a cerebral CT scan. Furthermore, if a carotid plaque <70% of stenosis was present, the DUS follow-up was mainly chosen, while in case of stenosis >70% a CT angiography of the carotid axis was performed to confirm the degree of stenosis and the possible indication for surgery (carotid endarterectomy or carotid artery stenting). The collected data were retrospectively analyzed in September 2022. The reported biomarkers were analyzed at baseline and every three months. The median follow-up was 19.2 (mean: 24; r: 12–51; standard deviation [SD]: 6.5) months. For statistical analysis, means and SD or median and range were reported for parametric data; absolute values and percentages were reported for non-parametric data. Statistical significance was considered at *p* < 0.05. Statistical analysis was performed using SPSS 16.0 (SPSS Inc., Chicago, IL, USA). 

### Laboratory Parameters and Biomarkers

Laboratory analyses included measuring serum levels of homocysteine, C-reactive protein (C-RP), and oxidized LDL. These findings were analyzed and recorded from routine blood tests.

## 3. Results

The study included 341 patients. The mean age was 53.8 (IQR: 33–87) years and 243 (71.3%) were male. Nonanatomic patient variables with the related grading system are reported in Table 1.

Among the 341 patients enrolled in the study, 100 (29.3%) had a thickness from the intimal–luminal to the medial–adventitial interface of <1.5 mm, below the value of definition of carotid plaques according to the Mannheim consensus [14]. A carotid plaque was diagnosed in 241 patients (70.7%), and in 62 cases (18.2%) a hemodynamic lesion with PSV greater than 120 cm/s was present (Table 2 and Table 3).

The cardiology visit performed on 241 included patients in the low-, medium-, and high-risk groups revealed a cardiac disease in 29.7% of patients. The incidence of cardiac disease in the low-, medium-, and high-risk groups was, respectively, 21.6% (6.1% of the total), 49% (7.3% of the total), and 60.2% (16.4% of the total), with a significantly higher incidence in patients in the high-risk group (*p* < 0.001). Fifteen patients (4.4%) were found positive for previous myocardial infarction with two cases of a silent heart attack in diabetic subjects. During the follow-up, a positive response for the symptomatic cardiologic disease was reported in 48 patients; 41 (12%) cases (10 medium grade, 31 high grade) indicating a coronarography requiring angioplasty; 7 (2%) cases (2 medium grade, 5 high grade) presenting a cardiac surgery indication. An asymptomatic stroke was reported in 20.8% (71) of patients. No other neurologic symptomatic events were observed during the follow-up. All patients with carotid plaques underwent peripheral DUS, diagnosing a PAD in 56.6% (193) of cases (Table 4). The clinical presentation of PAD was heterogeneous: in 13% (25 patients) PAD was asymptomatic; in 72% (138 patients) PAD was symptomatic with claudication (32 patients classified in Rutherford 1, 44 in Rutherford 2, and 62 in Rutherford 3); and in 15% (29 patients) PAD was symptomatic with critical limb ischemia (19 classified in Rutherford 4, and 10 in Rutherford 5).

All patients with plaques presented moderate levels of homocysteine, C-RP, and oxidized LDL with an improvement in mean value in all groups according to CAAD increase (Table 5). 

After the CAAD diagnosis, the best medical treatment with ASA, statins, L-Metilfolato (2 mg), vitamin B2 (7 mg), vitamin B6 (7 mg), and vitamin B12 (12.5 mcg) was administered in all patients. During follow-up, a basic sightseeing was performed every three months and a decrease in main vascular risk factors was observed. However, after six months, not all patients in the group of high-grade CAAD were responsive to therapy (31–33.3%; *p* < 0.05), with high levels of homocysteine, C-RP, and oxidized LDL during follow-up (Table 6). 

The difference of biomarker values in the high-grade CAAD group was statistically significative (*p* < 0.05). Therefore, in this “not responsive group” an elevated number of CT brain positive (27–87.2% *p* < 0.05) was present (Table 7). 

No death was reported in the follow-up, with interruption of checks in 27 patients (8%).

## 4. Discussion

In the present study, we evaluated the association of DUS biomarkers and serum biomarkers, highlighting the relationship between CAAD, coronary disease, and PAD. Carotid and coronary disease are considered part of a systemic atherosclerotic process and are therefore believed to share common risk factors [15,16,17,18,19]. About 85% of ictus cases are due to carotid artery stenosis, often asymptomatic before neurologic events [2,20,21]. Thromboembolism is the underlying cause in 40% of ischemic strokes with two-thirds of patients having ≤50% stenosis for carotid artery disease [22,23,24]. 

Our study shows a correlation between the progression of CAAD and cardiologic risk. Patients with atherosclerotic plaques have shown a higher rate of cardiologic procedures after stenosis diagnosis, despite the asymptomatic condition reported at baseline. Therefore, the greater the degree of CAAD, the greater the risk was of coronary disease involving symptomatic events. Statistically significative (*p* < 0.05) was the difference in the response rate of biomarkers between advanced CAAD and the other groups after the best medical therapy. These data are consistent with the literature, where the surgical treatment is indicated in advanced CAAD as the result of the reduced efficacy of medical therapy [25,26]. Different studies in the literature suggest that elevated blood pressure, smoking, cholesterol levels, increasing age, and male sex are associated with the presence of carotid artery stenosis. These risk factors were significatively present in the high-grade group, where 67% were responsive to therapy. The remaining group, with the absence of improvements overall in the cases with a cerebral lesion, presents different and interesting insights. 

In the literature, there is no agreement about the treatment of symptomatic patients because CEA procedures have shown no difference in ipsilateral stroke rates after 6-months to 1-year post-CEA compared to the sole preferred medical treatment. Consequently, European guidelines suggest that patients who do not present with clinical symptoms (or plaque vulnerability factors) should not undergo CEA [2]. More recently, a significant reduction in complications after CEA and CAS was reported, due to improvements of both surgical techniques and sorting systems for high-risk subjects [27,28]. The increase in serum biomarkers according to the worsening of DUS biomarkers could be used further as discriminating factors in the therapeutic path to allow the best choice for the patient. 

However, viewing the carotid artery as a biomarker, an expression of the quality of our arterial system, is a fascinating and at same time interesting concept. The possibility to resort to noninvasive diagnostic tools and not expansive serum biomarkers is a focal point for reflection to encourage the efficiency of preventive medicine. Kurosaki et al. reported a higher risk of stroke in a patient with ulcerated plaque and in a diabetic patient, suggesting an improvement in the prediction model of a future ischemic event due to the study integration of luminal stenosis, plaque composition, and mechanical loading [29]. Zhou KN et al. found carotid artery strain to be a simple and promising indicator for cardiac dysfunction and to help in the clinical grading and risk stratification of heart failure [30]. Our data show an increase in serum biomarkers during follow-up with greater speed in the group with more significative plaques, despite the absence of significative static variations. 

Interesting data from our experience show a higher stroke risk for patients not responsive to therapy, where serum biomarkers do not decrease during follow-up and under the best therapy. The possible mechanisms explaining the failure to improve biomarker levels are not yet fully established. The main hypotheses based on the literature are that elevated homocysteine, C-RP, and oxidized LDL levels may have enhanced oxidative stress and inflammation of vascular endothelial cells that in cases of severe atherosclerosis have deep damage, not curable with the best therapy [31,32,33].

In this study, the assessment of hemodynamically significant stenosis was completed with a cerebral CT scan and a CT angiography in the case of stenosis >70% in order to confirm and to objectively evaluate the plaque and its features. In the literature and in our experience, in addition to the degree of luminal stenosis, the plaque composition is also essential to evaluate and discuss. From mild fatty deposition in asymptomatic patients to complex, irregular plaques prone to the thromboembolism, DUS, CT scan, and high-resolution MR can provide significant information [34,35]. DUS is useful in determining hemodynamic features; Choi E. et al. report how CT scans help to assess the composition of plaque, with Hounsfield unit density and spotty calcium considered independent predictors of a greater risk of adverse cardiovascular event occurrence; Porambo ME et al. underline the ability of MR in the study of vulnerability plaque showing thoroughly lipid-rich necrotic core, thin/ruptured fibrous cap, and intraplaque hemorrhage, predicting factors of future stroke [36,37]. However, every imaging method can be a source of information about carotid plaques with DUS representing the first line of approach to CAAD, and CT scans and MR are useful tools to identify patients with high risk [38].

The presence of PAD and its close connection with CAAD and coronary disease is another element that emerges in this analysis. In the literature, severe PAD finds a high rate of coronary artery disease [39]. This item is important because, unlike CAAD and coronary disease which can be asymptomatic for long time, PAD can present early clinical signs with a fast path to the main diagnosis. In contrast to other vascular diseases, in this experience, PAD did not show a progressive increase parallel to the worsening of CAAD with the spreading of the different degrees of peripheral disease. In the literature, this match is common; in fact, in terms of cardiology risk more relevance is given to multilevel vascular disease than that involving the singular district [40,41,42,43]. 

The message from the literature is clear, there is a 24.5% incidence of a >50% internal carotid artery stenosis or occlusion on duplex examination in patients with claudication without cerebrovascular symptoms [44,45,46]. Polyvascular patients should undergo routine multi distrectual DUS screening to detect asymptomatic vascular disease.

## 5. Conclusions

The reported study shows that ultrasound biomarkers and serum biomarkers can be used simply to follow asymptomatic patients with an evolutive risk of stroke. The increase in serum biomarkers and the worsening of DUS biomarkers could be used to identify patients with a high risk of stroke. Atherosclerosis is a multilevel vascular disease, and the study of a particular vascular district should not exclude others. Ultrasound parameters and a multiple biomarkers approach can be employed to identify early patients at higher risk of disease progression or inefficient therapy. The frequent multilevel localization of CAAD, coronary arteries disease, and peripheral disease should be aggressively managed despite the higher risk of comorbidities and complications. A standardized approach to main vascular disease is necessary to improve outcomes in the prevention of stroke events.

## Figures and Tables

**Figure 1 diagnostics-13-00644-f001:**
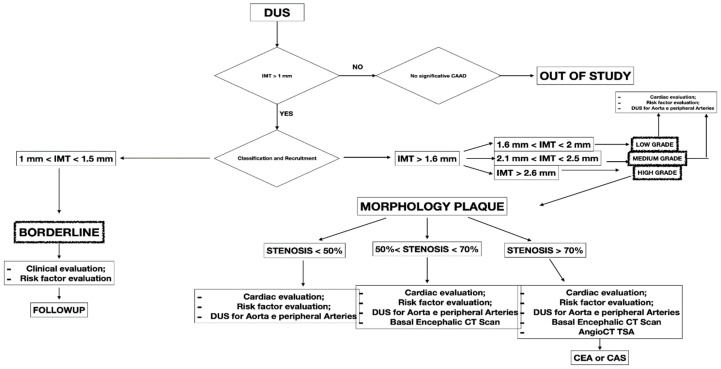
Flow chart showing the decisional algorithm employed in patients presenting with carotid artery disease (CAAD).

**Figure 2 diagnostics-13-00644-f002:**
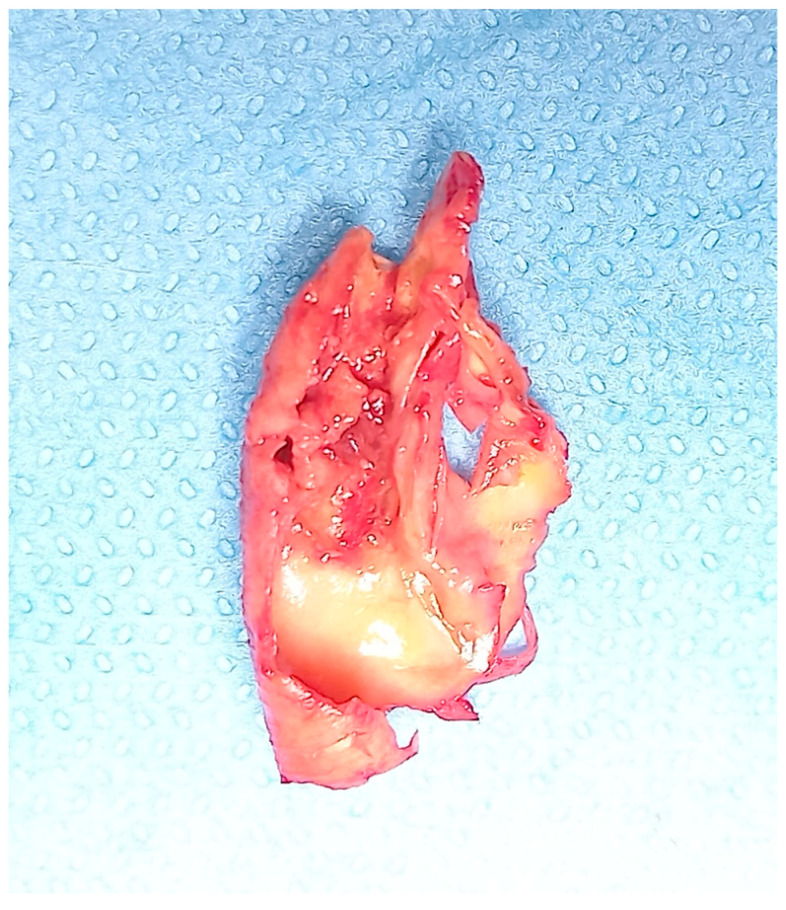
Ulcerated carotid plaque after endarterectomy.

**Figure 3 diagnostics-13-00644-f003:**
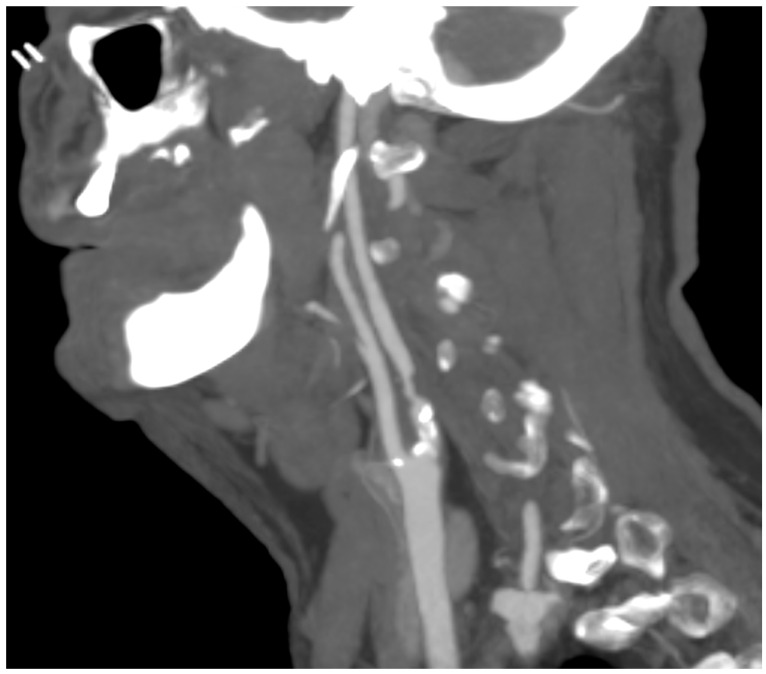
CT angiography showing carotid plaque with significative stenosis.

**Figure 4 diagnostics-13-00644-f004:**
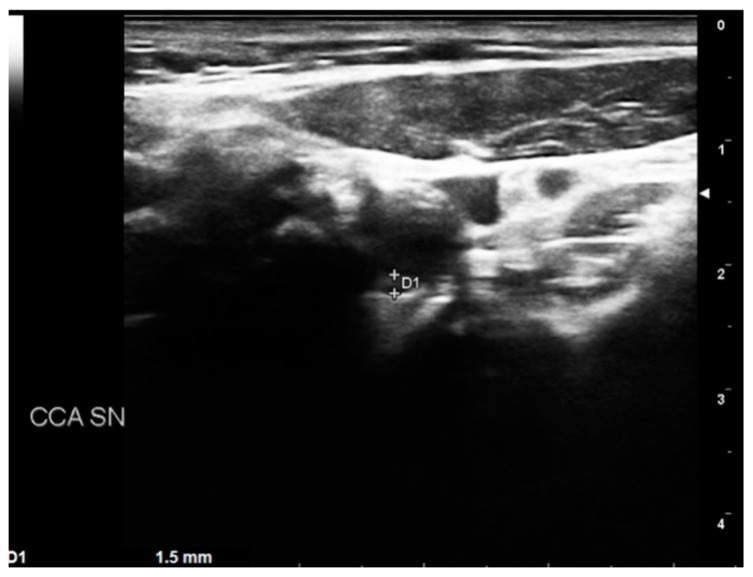
DUS image showing plaque in CCA—BORDERLINE.

**Figure 5 diagnostics-13-00644-f005:**
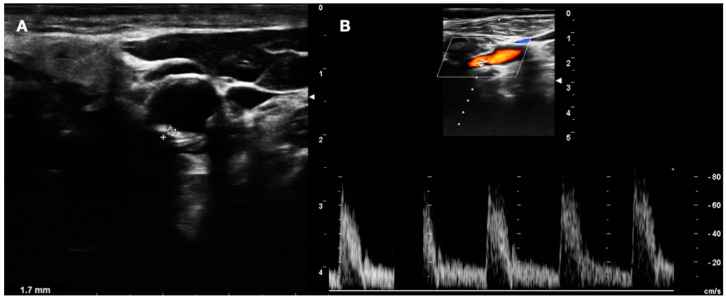
DUS images showing plaque in CCA—LOW GRADE (**A**), and its relative flow (**B**).

**Figure 6 diagnostics-13-00644-f006:**
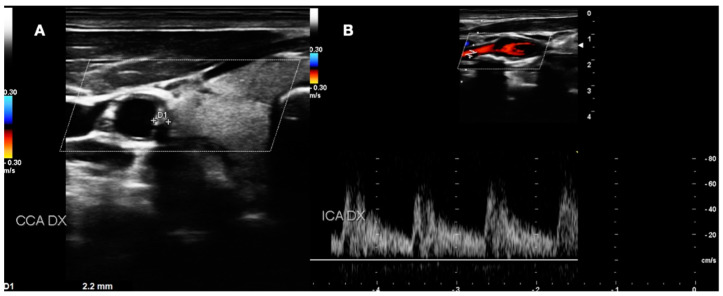
DUS images showing plaque in CCA—MEDIUM GRADE (**A**), and its relative flow (**B**).

**Figure 7 diagnostics-13-00644-f007:**
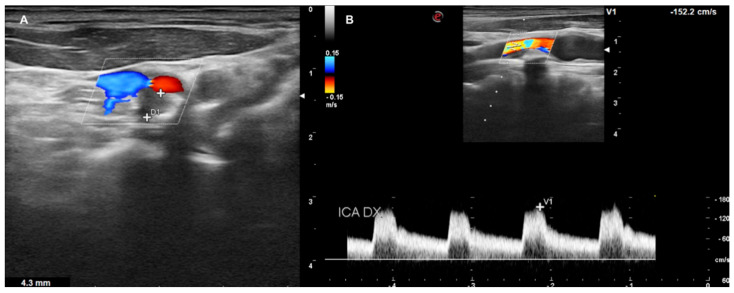
DUS images showing plaque in CCA—HIGH GRADE (**A**), and its relative flow (**B**).

**Table 1 diagnostics-13-00644-t001:** Nonanatomic patient variables.

Categories	Grade	N	%
Diabetes	None	169	49.6
	Not requiring insulin	95	27.8
	Controlled by insulin	75	22
	Type 1 or uncontrolled	2	0.6
Tobacco Use	None (>10 years ago)	144	42.2
	Quit 1–10 years ago	45	13.2
	Current within last year, <1 package per day	113	33.1
	Current within last year, >1 package per day	39	11.5
Hypertension	None	66	19.3
	Controlled with 1 drug	78	22.9
	Controlled with 2 drugs	138	40.5
	Requiring >2 drugs or uncontrolled	59	17.3
Renal Status	Normal	222	66.1
	Evidence of renal disease, GFR > 90 mL/min/1.73 m^2^	55	15.1
	GFR 60–89 mL/min/1.73 m^2^	24	7.1
	GFR 30–59 mL/min/1.73 m^2^	14	4.1
	GFR 15–29 mL/min/1.73 m^2^	12	3.5
	GFR < 15 mL/min/1.73 m^2^	14	4.1
Cardiac status	Asymptomatic	234	69
	Asymptomatic, but with remote myocardial infarction by history (6 months)	89	26
	Stable angina, ejection fraction 25% to 45%, controlled ectopy, or history of congestive heart failure that is now well compensated	18	5
	Unstable angina, ejection fraction <25%, myocardial infarction ≤6 months	0	0

**Table 2 diagnostics-13-00644-t002:** Classification of patients with CAAD.

Categories	Lesion		N	%
Borderline	1 mm < IMT < 1.5 mm			
			100	29.5


Low Grade	1.6 mm < IMT < 2 mm		97	28.4
Medium Grade	2 mm < IMT < 2.5 mm		51	
				14.9

High Grade	IMT > 2.6 mm	Stenosis < 50% (I)	21	6.1
		50% < Stenosis < 70% (II)	35	10.3
		Stenosis > 70% (III)	37	10.8

**Table 3 diagnostics-13-00644-t003:** Classification of plaques.

Categories	Variable	N	%
Low Grade	Smooth	97	100
	Irregular	0	0
	Ulcerated	0	0
	Soft	0	0
	Calcified	0	0
Medium Grade	Smooth	34	10
	Irregular	17	5
	Ulcerated	0	0
	Soft	44	13
	Calcified	7	2
High Grade (I)	Smooth	11	3.2
	Irregular	8	2.3
	Ulcerated	2	0.6
	Soft	15	4.4
	Calcified	6	1.7
High Grade (II)	Smooth	10	3
	Irregular	22	6.4
	Ulcerated	3	0.9
	Soft	14	4.1
	Calcified	21	6.1
High Grade (III)	Smooth	13	3.8
	Irregular	20	6
	Ulcerated	4	1.2
	Soft	21	6.1
	Calcified	16	4.7

**Table 4 diagnostics-13-00644-t004:** Classification of Stenosis.

Categories	Variable	N	%	VPS (cm/s)
Low Grade	Carotid Stenosis Degree		20 +/− 5	77 +/− 2
	Stroke	0	0	
	Cardiac Disease	21	6.1	
	PAD	67	19.6	
Medium Grade	Carotid Stenosis Degree		33 +/− 4	78 +/− 8
	Stroke	2	0.6	
	Cardiac Disease	25	7.3	
	PAD	48	14.1	
High Grade (I)	Carotid Stenosis Degree		42 +/− 5	88 +/− 10
	Stroke	3	0.9	
	Cardiac Disease	12	3.5	
	PAD	25	7.3	
High Grade (II)	Carotid Stenosis Degree		55 +/− 2	133 +/− 5
	Stroke	3	0.9	
	Cardiac Disease	21	6.1	
	PAD	28	8.2	
High Grade (III)	Carotid Stenosis Degree		80 +/− 2	220 +/− 22
	Stroke	3	1.2	
	Cardiac Disease	23	6.7	
	PAD	25	7.3	

**Table 5 diagnostics-13-00644-t005:** Serum Biomarkers.

Categories	Variable	Vm	Vm6
Borderline	Serum Homocisteine (µmol/L)	10.7 +/− 2.1	9.5 +/− 1.5
	Ox-LDL (µg/L)	<235	<235
	C-reactive protein (mg/dL)	0.7 +/− 0.1	0.6 +/− 0.1
Low Grade	Serum Homocisteine (µmol/L)	11.8 +/− 2.0	10.7 +/− 1.1
	Ox-LDL (µg/L)	251 +/− 1.9	244 +/− 1.5
	C-reactive protein (mg/dL)	1.7 +/− 0.2	1.5 +/− 0.2
Medium Grade	Serum Homocisteine (µmol/L)	11.8 +/− 2.2	11.4 +/− 1.4
	Ox-LDL (µg/L)	253 +/− 1.7	249 +/− 0.4
	C-reactive protein (mg/dL)	2.1 +/− 0.5	1.9 +/− 0.7
High Grade (I)	Serum Homocisteine (µmol/L)	12.3 +/− 2.1	12.2 +/− 2.0
	Ox-LDL (µg/L)	260 +/− 1.7	262 +/− 1.5
	C-reactive protein (mg/dL)	2.9 +/− 0.2	2.7 +/− 0.1
High Grade (II)	Serum Homocisteine (µmol/L)	12.1 +/− 1.4	12.8 +/− 1.8
	Ox-LDL (µg/L)	270 +/− 2.1	275 +/− 1.7
	C-reactive protein (mg/dL)	3.2 +/− 0.2	3.4 +/− 0.3
High Grade (III)	Serum Homocisteine (µmol/L)	12.7 +/− 1.1	13.5 +/− 1.2
	Ox-LDL (µg/L)	275 +/− 2.3	280 +/− 0.3
	C-reactive protein (mg/dL)	3.9 +/− 0.9	4.1 +/− 0.2

**Table 6 diagnostics-13-00644-t006:** Serum biomarkers in high-grade group.

Categories		Variable	Vm	Vm6
High Grade	RESPONSIVE	Serum homocisteine (µmol/L)	12.2 +/− 1.1	11.9 +/− 0.8
	NO RESPONSIVE	Serum homocisteine (µmol/L)	12.7 +/− 1.3	13.2 +/− 1.9
	RESPONSIVE	Ox-LDL (µg/L)	258 +/− 1.7	248 +/− 1
	NO RESPONSIVE	Ox-LDL (µg/L)	265 +/− 1.4	274 +/− 1.6
	RESPONSIVE	C-reactive protein (mg/dL)	2.7 +/− 0.1	2.5 +/− 0.5
	NO RESPONSIVE	C-reactive protein (mg/dL)	3.1 +/− 0.3	3.4 +/− 0.1

**Table 7 diagnostics-13-00644-t007:** Distribution of CT brain positive.

Categories	N	CT Positive	%	%T
Borderline	100	0	0	0
Low Grade	97	0	0	0
Medium Grade	51	1	2%	3.2%
High Grade	21	2	9.5%	6.4%
	35	1	2.8%	3.2%
	37	27	73%	87.2%

## Data Availability

The data presented in this study are available on request from the corresponding author.

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
