# Peer review of "Carotid Artery Disease in the Era of Biomarkers: A Pilot Study"

_diagnostics, 2023, doi:10.3390/diagnostics13040644_

Round 1

Reviewer 1 Report

Interesting and well written study regarding the Carotid Artery Disease in the era of Biomarkers.

Abstract: I would suggest to specify which serum biomarkers can be used to assess the degree of atherosclerotic disease and cardiovascular risk; furthermore al the end of the abstract you could also mention which biomarkers you used in the "multiple biomarkers approach" to be effective to identify early patients at higher risk of disease progression or inefficient therapy. These aspect can be useful to be more direct with potential readers and to be more specific for readers who won't read all the article, but just the abstract.

Introduction: I would add in this section that since the beginning of endovascular treatment of acute stroke (2015) in common clinical practice to treat acute stoke due to common/internal carotid artery pathology (eg: Sallustio F, et al. Carotid artery stenting during endovascular thrombectomy for acute ischemic stroke with tandem occlusion: the Italian Registry of Endovascular Treatment in Acute Stroke. Acta Neurol Belg. 2022 Sep 2. doi: 10.1007/s13760-022-02067-z. Epub ahead of print. PMID: 36056270.), above all since complication regarding this treatment are low (see for example: Salsano Get al. Complications of mechanical thrombectomy for acute ischemic stroke: Incidence, risk factors, and clinical relevance in the Italian Registry of Endovascular Treatment in acute stroke. Int J Stroke. 2021 Oct;16(7):818-827. doi: 10.1177/1747493020976681. Epub 2020 Dec 6. PMID: 33283685.)

M&M and rewsults: nothing to concern about. 

Discussion: you could add some lines for discussing the different role of CT or MR, in the definition of carotid artery disease, and compare those aspects to US and biomarkers. 

Conclusions: ok.

Author Response

Thanks for your review. Following your instructions, I improved the paper

Reviewer 2 Report

Reviewer is vascular neurosurgeon and load of pts to CEA is now lower and lower… Indication for surgery is in this pharmacological era difficult  and  combination of sonographic results and level of biomarkers and their dynamics  is promising.

CEA is now very safe procedure do to neurosurgical technology and  perioperative monitoring. Group of pts with benefit from surgery but still exist and correct and exact selection is mandatory.  From   this point  good job, well done and  recommended to publishing.

Author Response

Thanks for review. I improved my paper
